# Targeted Antibacterial Endolysin to Treat Infected Wounds on 3D Full-Thickness Skin Model: XZ.700 Efficacy

**DOI:** 10.3390/pharmaceutics16121539

**Published:** 2024-12-01

**Authors:** Marisa Meloni, Bob de Rooij, Ferdinand W. Janssen, Francesca Rescigno, Bernadette Lombardi

**Affiliations:** 1VitroScreen s.r.l., In Vitro Innovation Center, Via Mosè Bianchi 103, 20149 Milan, MI, Italy; marisa.meloni@icloud.com (M.M.); lombardi.bernadette@gmail.com (B.L.); 2Micreos Pharmaceuticals, Neuhofstrasse 12, CH-6430 Baar, ZG, Switzerland; b.derooij@micreos.com (B.d.R.); f.w.janssen-9@prinsesmaximacentrum.nl (F.W.J.)

**Keywords:** wound healing, re-epithelization process, anti-microbial effect, injury model, anti-microbial adhesion, biofilm efficacy

## Abstract

**Backgrounds/Objectives:** Skin wound healing is a physiological process orchestrated by epithelial and mesenchymal cells able to restore tissue continuity by re-organizing themselves and the ECM. This research study aimed to develop an optimized in vitro experimental model of full-thickness skin, to address molecular and morphological modifications occurring in the re-epithelization and wound healing process. **Methods:** Wound healing starting events were investigated within an experimental window of 8 days at the molecular level by gene expression and immunofluorescence of key epidermal and dermal biomarkers. To mirror the behavior of infected wounds, the established wound healing model was then colonized with *S. aureus*, and the efficacy of a novel antibacterial agent, XZ.700, was investigated. Viable counts (CFU/tissue), IF, and ultrastructural analysis (SEM) were performed to evaluate *S. aureus* colonization inside and around the wound bed in an experimental window of 3 h of colonization and 24 h of treatment. **Results:** Endolysin showed an efficacy in counteracting bacterial growth and invasion within the wound bed, reducing the *S. aureus* load compared to its placebo, thanks to its selective antimicrobial activity interfering with biofilm formation. **Conclusions:** The preclinical in vitro infected wound model on FT-kin showed interesting applications to assess the repair efficacy of dermo-pharmaceutical and cosmetic formulations.

## 1. Introduction

The skin functions as a physical barrier against the environment and plays a primary role in protecting the body from external influences, such as xenobiotics, pathogenic microorganisms, and injuries [1,2,3]. This defense mechanism is guaranteed by both cell–cell interaction in the epidermis and a tightly controlled balance between skin cells, growth factors, and signaling cascades [4,5,6]. The impairment of skin integrity, as in the case of a wound [7], pathological modifications, or the accumulation of advanced glycation end products (AGE) in diabetes [8], compromises this defense mechanism. In this *scenario*, different cells collaborate to restore the lesion induced by a local aggression, beginning with a very early inflammatory stage [3,9].

The skin repair process is ensured by keratinocytes and fibroblasts, which play a predominant role in ensuring the integrity and main skin functions [10,11]. Epidermal cells initially work to quickly re-establish the barrier function by activating re-epithelialization; subsequently, activated fibroblasts induce extra-cellular matrix (ECM) deposition and remodeling [12,13]. Under physiological conditions, the starting point of the skin repair process is the disruption of tissue continuity, which stimulates the migration and proliferation of keratinocytes in the wound bed, with the subsequent deposition of collagen and a new matrix by fibroblasts [14]. In particular, during the re-epithelialization, multiple cellular and molecular processes are activated, firstly allowing keratinocytes to migrate from the wound edges to generate a provisional wound bed matrix, where they can proliferate, differentiate, and stratify into a new epidermis [12,15,16,17]. The keratinocytes migration allows fibroblasts activation, proving a provisional ECM rich in fibronectin (FN) and hyaluronic (HA). This provisional matrix is then replaced with new collagen and a mature ECM [18,19,20,21].

The ability of cells to re-epithelialize is a key indicator of wound healing [22]. The scratch assay is a simple and affordable method to study cell migration on a 2D surface. It is widely used in drug discovery for initial screening, due to its ease of standardization and automation [23,24]. However, 2D cultures lack the complexity of 3D tissue environments, where cell–cell and cell–matrix interactions play a crucial role. This can lead to inaccurate predictions of the in vivo response [25,26].

To address this limitation, more sophisticated 3D cell culture models have been developed [27]. Several methods have been developed to induce wounds, including burn injuries, frostbite, and excisional and incisional wounds [28,29,30,31,32,33]. While these models provide a more physiologically relevant environment, they are more complex and time-consuming to analyze [25]. Unlike the simple gap closure measurement in scratch assays, evaluating wound healing in 3D models requires histological techniques, similar to analyzing real skin tissue. This allows for a more comprehensive assessment, including the formation of the basement membrane and the quality of the newly synthesized extracellular matrix [34].

In addition to the re-epithelization process, the cutaneous flora could exhibit a critical role in the skin disorders and pathological conditions. It is estimated that microorganisms live on the human skin surface at ~10^6^ cells/cm^2^, principally made up of coryneform bacteria (*Corynebacterium*, *Brevibacterium*, and *Propionibacterium*) and *staphylococci* [35,36], all in a physiological homeostatic equilibrium. The unbalancing of the physiological microbial charge could allow for the critical proliferation of endogenous/exogenous microorganisms, and trigger more or less extensive infectious disorders [36]. Several studies have demonstrated that physicochemical interactions’ surfaces or topography may also be implicated in microbial adhesion [37], proliferation, and biofilm formation, playing a critical role in infected skin lesions [38]. Indeed, the formation of biofilm (rich polysaccharides, proteins, and lipids) represents one of the most severe complications of microbial infections, allowing microorganisms’ survival also in hostile environments [39,40,41,42]. The onset of biofilms is often associated with the failure of the physiological wound healing process, leading to a stalled chronic condition [42,43]. In particular, *Staphylococcus aureus* is a common bacterial pathogen that frequently infects these wounds, leading to persistent infections, delayed healing, and potentially serious complications such as amputation. The effective management of diabetes wounds requires early diagnosis, appropriate antimicrobial therapy, and accurate care to prevent infection and promote healing [44,45].

In this *scenario*, an in vitro experimental system based on reconstructed human skin able to mirror the infected skin lesions (wound), the dynamic of bacterial infection, and its impact on skin is long overdue. Advanced tissue engineering techniques are applied for the novel generation of tissue-engineered skin models that closely resemble normal human skin.

Within a common research project, an in vitro experimental model on reconstructed full-thickness skin tissue was developed, which addresses molecular and morphological modifications that occur in the wound healing process and in the presence of *S. aureus*, to validate the efficacy of endolysin XZ.700 in counteracting bacterial growth, invasion within the wound bed, and biofilm formation.

Endolysins are a class of antibacterial enzymes encoded by bacteriophages. In nature, they degrade the peptidoglycan bacterial cell wall at the end of the phage replication cycle, and hereby kill bacteria and cause the release of progeny bacteriophages [46]. Endolysins are currently being developed as a novel class of antimicrobials because of their rapid killing kinetics, the high level of specificity for their target cells, their activity against drug-resistant bacteria and biofilms, and their low chance of inducing bacterial resistance due to their highly conserved target sites in the Proteoglycan [47,48]. In this paper, we investigated the potency of the chimeric endolysin called “XZ.700”, which is composed of functional domains from the bacteriophage endolysin Ply2638 and the bacteriocin lysostaphin [49], designed to specifically target *S. aureus* when applied topically. The concentration of XZ.700 applied was defined on the basis of pre-clinical studies performed using animal and human skin models [49].

The Phenion^®^ Full-Thickness Skin Model (FT-skin), produced by Henkel (Düsseldorf, Germany), is a 3D human skin equivalent reconstructed with air lift technology using human cells. It has previously been demonstrated to be suitable to simulate wound lesions and studying interactions between the epidermis structure. Before microbial inoculum, the FT-skin model epithelial integrity was challenged by performing a specific lesion (wound) using a standardized procedure. Once injured, the FT-skin models were colonized with *S. aureus* and exposed to XZ.700 endolysin (Micreos Pharmaceuticals, Baar, Switzerland) to evaluate its impact on bacterial growth, localization, and phenotype during the first 24 h after injury. The microenvironment around the wound was investigated by immunofluorescence (IF) in multiple visualizations (on top-view images and on transversal sections), giving unique insights on the bacterium interaction with host cells, confirmed by an ultrastructural analysis provided by scanning electronic microscopy.

The main effect of specific endolysin treatments is a direct activity on the *S. aureus* strain as a planktonic bacterium, in counteracting biofilm phenotype formation and promoting wound healing compared to its placebo formulation.

Taken together, all the results confirmed the efficacy of endolysin on the *S. aureus* contamination that occurred in infected skin lesions, paving the way for the new generation of innovative topical treatments thanks to the use of advanced engineered 3D skin models; their high biological relevancy allows a mechanistic understanding of the impact of active molecules on a living tissue, better mirroring human exposure.

## 2. Materials and Methods

### 2.1. Test System

The Phenion^®^ FT Skin Models (FT-skin) are artificial human 3D models composed of both epidermis and dermis, produced by Henkel (Düsseldorf, Germany), with applications in dermo-cosmetics and toxicology fields [50,51,52,53]. The FT-skin models are produced from cells of single donors, and they characterized by a diameter of 1.4 cm and a tissue surface area of 1.5 cm^2^. The models are generated by validated cells, tested for the absence of HIV-1/-2, Mycoplasma, and Hepatitis B and C. The Full-Thickness models are cultured in an air–liquid interface with a specific growth medium provided by the supplier. Tissues are cultured in standard conditions (37 °C, 5% CO_2_, and 90% RH).

### 2.2. Test Item

XZ.700, produced by Micreos Pharmaceuticals (Baar, Switzerland), is a non-antibiotic treatment strategy aimed at controlling *Staphylococcus aureus* (*S. aureus*) colonization of lesional skin. This product is a recombinant bacterial cell wall hydrolase composed of functional domains from the bacteriophage endolysin Ply2638 and the bacteriocin lysostaphin [49]. It is formulated in a 1.2% HPMC gel matrix. The efficacy dose chosen was 300 μg/g of active compared to placebo, with 0 μg/g of active added to the final formulation.

### 2.3. Skin Injury

FT-skin tissues were transferred in sterile conditions in a Petri dish, and wounded with a sterile punch biopsy equipped with a plunger of 1 mm diameter (Kai Industries Corporation, Seki City Gifu, Japan). The 1 mm punch biopsy allows a core drilling, by removing a defined cylinder of tissue to create a physical vertical lesion in both the epidermis and dermis layers, able to induce the activation of the repair process.

### 2.4. Quantitative Real-Time PCR

For the evaluation of the re-epithelization process, FT-skin models were collected in the RNaqueous Lysis buffer (Thermofisher, Waltham, MA, USA), and the total RNA was extracted using the column-based RNaqueous kit (Thermofisher, Waltham, MA, USA) and spectrophotometrically quantified. The RNA quality was evaluated by electrophoresis on agarose gel. cDNA was produced by retro-transcription with High Capacity Retrotranscriptase (Thermofisher, Waltham, MA, USA) and used for gene expression analysis by quantitative PCR in real time, using the TaqMan Master Mix (Thermofisher, Waltham, MA, USA) and the following Taqman assay primers and probes: GAPDH *Hs_99999905_m1*, *TGF-β1 Hs_00998133_m1*, and *KFG (FGF-7) Hs_00384281_m1* (Thermofisher, Waltham, MA, USA).

A relative quantification (RQ study) analysis was performed following the ΔΔCt calculation method within a 40-cycles reaction (Ct max = 40). The relative quantification (RQ) value was calculated using a negative control as the calibrator and the expression of GAPDH as the calibrator gene for data normalization. The internal instrument level of confidence used was 95%. Signal quantification was performed in comparison with a calibrator series (in this case, a non-wounded tissue). For the data analysis, the relative quantification value (RQ) was accepted as significant when the gene was “one-fold” up- (RQ > 2) or down-regulated (RQ < 0.5) compared to the calibrator sample (RQ = 1).

### 2.5. Histology and Immunohistochemistry Analyses

All wounded tissues were collected for immuno-histological analyses at defined timepoints; FT-skin models not wounded were used as a control.

Tissues were fixed in a 10% formalin-buffered solution (Merck, Darmstadt, Germany), included in paraffin blocks, and sectioned in 5 μm slides. Tissue sections were stained with Hematoxylin and Eosin (H&E) (Histoline, Pantigliate, Italy) and immune-labeled for the following biomarkers: Cytokeratin 10 (CK10, 1:200, rabbit polyclonal antibody, Merck, Darmstadt, Germany), Cytokeratin 14 (CK14, 1:200, mouse monoclonal antibody, Abcam, Cambridge, UK), Integrin-β1 (ITG-β1 1:50, mouse monoclonal antibody, Abcam, Cambridge, UK), α-smooth muscle actin (α-SMA 1:100, mouse monoclonal antibody, (Abcam, Cambridge, UK), and FN (1:100, rabbit polyclonal antibody, Merck, Darmstadt, Germany). Alexa Fluor 555 donkey anti-rabbit and Alexa Fluor PLUS 488 Got anti-mouse (Invitrogen, Waltham, MA, USA) were used as secondary antibodies diluted 1:400 in PBS.

DAPI (Merck, Darmstadt, Germany) was used for nuclei staining. The images were acquired with the LEICA DMi8 THUNDER imager 3D system (Leica, Wetzlar, Germany). Postprocessing imaging was performed by ImageJ software (version: 2.14.0/1.54f) and the Leica Tool “Analyse” on LASX software (version: 3.7.5, Leica, Wetzlar, Germany), and the signal of expression was normalized to the cell number. Cell density was calculated by dividing the number of cells counted per area using a well-defined region of interest (ROI = 100 × 100).

To proceed to the top-view and cross-sections immunolabelling for *S. aureus*, a different procedure was applied as follows: Phenion^®^ FT-skin was cut with a scalpel near the injured area and the half tissue comprising the lesion was placed in the histo-cassette, fixed in a 10% formalin-buffered solution (Merck, Darmstadt, Germany) overnight at +4 °C and then embedded in paraffin blocks. Tissue slices of 5 μm were obtained and immune-labeling for *S. aureus* was performed by the anti-*S.aureus* primary antibody (Abcam, Cambridge, UK). The signal was visualized with Alexa Fluor 555 PLUS (Invitrogen, Waltham, MA, USA) and nuclei were counter-stained with DAPI (Merck, Darmstadt, Germany).

### 2.6. Bacterial Strain and Inoculum

The MRSA *S. aureus* bacterial strain (ATCC 33591) was thawed, inoculated onto culture medium (BHI agar plate), and incubated at 37 °C in aerobic conditions for at least 24 h in order to obtain a fresh culture. Once the growth started, 2 passages before the colonization were performed. On the day of the experiment, bacterial strain was resuspended in sterile saline solution at the required concentration range. The bacterial inoculum concentration was checked by the spectrophotometric OD measure and adjusted to obtain a range of about 1 × 10^8^ CFU/mL (OD = 1), mirroring a severe infection. A volume of 10 µL of *S. aureus* suspension with the inoculum level of 1 × 10^8^ CFU/mL (equivalent to 10^6^ CFU/tissue) was added to the tissue surface precisely around the injury site with a micropipette, avoiding touching the epidermal surface. The colonized tissues were cultured in standard conditions in an incubator at 37 °C, 5% CO_2_, and 90% RH.

A viability check of the colonization inoculum was performed at T0 on a BHI agar plate by the spread method, using different 10-fold dilutions performed in saline solution and incubated at 37 °C (aerobic conditions) for 24–48 h.

### 2.7. Bacterial Viable Count on Tissue Homogenates and XZ.700 Treatment

The Phenion^®^ FT-skin was treated with 20 µL of the test item (XZ.700 300 µg/g) or placebo (XZ.700 0 µg/g), applied to the injury site with a micropipette (avoiding touching the epidermal surface) on the same area infected with the *S. aureus*. The tissues were incubated in standard conditions at 37 °C, 5% CO_2_, and 90% RH.

Tissues designated for viable counts (CFU/tissue) were collected in a 7 mL vial for homogenization, containing 1.5 mL of saline buffer solution and processed by Minilys Homogenizer (Bertin Technologies, Montigny-le-Bretonneux, France). The number of residual viable colonies of *S. aureus* was determined by serial decimal dilutions and the spread method on agar plates.

The following Formula (1) was used to calculate the CFU/mL:(1)$$\frac{CFU}{mL}=\frac{n{}^{\circ}\;of\;colonies\times DF}{0.1}\mathrm{mL}$$
where DF is the dilution factor applied. Then, all the results were converted to CFU/tissue.

### 2.8. Scanning Electron Microscopy (SEM)

For the ultrastructural analysis, the FT-skin designated for SEM (Nova NanoSEM 450, Fei Company, Hillsboro, OR, USA) was collected, covered with 2 mL of fixative buffer solution (based on glutaraldehyde solution 2%), placed into a 24-well plate, and stored at 2–8 °C until the analyses were performed.

### 2.9. Statistics

Statistical analyses on immunohistochemistry results were performed by an ANOVA-Tukey HSD test. The significance threshold was set for the Tukey HSD test as *p*-value < 0.01. Experiments were repeated in biological triplicates in 3 independent studies for all data sets.

## 3. Results

### 3.1. Wound Model (WM)

Skin lesions were generated by a punch biopsy equipped with a sterile plunger with a 1 mm diameter. To ensure a deep vertical skin lesion, core drilling was performed in order to remove a defined tissue cylinder composed of both epidermis and dermis tissue with a surgical precision (Figure 1).

### 3.2. Re-Epithelization Model: Dynamic Wound Healing in 8 Days

#### 3.2.1. Keratinocytes Migration in Response to Injury in WM After 1 and 3 Days

The keratinocytes migration was assessed during the early steps of the repair process (1–3 days), by the localization of CK14 and ITG-β1 in IF; furthermore, nuclei orientation in the wound bed of the WM after three days of post-injury culture was evaluated. The analyses were performed by comparing neo-formed tissue in the WM and the tissue of the NC.

Representative fluorescent images of CK14 and ITG-β1 in the negative control (NC) compared to the WM on day 1 and day 3 are reported in Figure 2 (green signal). The CK14 signal was homogeneously distributed throughout the epidermis in the NC tissue at day 1 (Figure 2A), whereas it was expressed only in the first two layers of the epidermis at day 3 (Figure 2B), indicating a reduced number of basal keratinocytes. Furthermore, the epidermis thickness decreased in cell numbers from day 1 to day 3, and the number of cells (Blue-DAPI stain) decreased by nearly 50% on day 3 (Figure 2B). In the injured series, the epidermal architecture in the wound bed on day 1 appeared unorganized and CK14-positive cells were nearly absent in the wound bed (Figure 2C). On day 3 (Figure 2D), CK14-positive keratinocytes were strongly detectable, with a fluorescence signal well-structured and organized in the wound bed of the WM tissue. CK14 was expressed in both the basal and upper layers of the newly formed epidermis in the wound bed, with an extensive distribution of cytokeratin in all areas.

Despite CK14, the ITG-β1 expression and localization appeared drastically different in the WM tissue during repair; ITG-β1 expression was visible in the basal layer of the epithelium in the wound bed on day 1 (Figure 2E), suggesting an activation of keratinocytes migration into the wound bed. On day 3 (Figure 2F), the reduction in expression signal indicated the rapid disappearance of the ITG-β1. These preliminary results suggest and confirm that migration occurred in the wound bed in the very early phase of the tissue repair process, within 3 days. Wound bed repopulation and time-dependent CK14 expression also suggest an early proliferation mechanism.

#### 3.2.2. Keratinocytes Nuclei Orientation in WM

Cellular re-organization was also assessed, by evaluating the nuclei orientation in the neo-formed tissue inside the wound area for signs of collective polarization, indicating a potential cell stream toward the wound (Figure 3). The nuclei orientation in the WM (Figure 3A_1_) and the NC (insert Figure 3A_2_) on day 3 are illustrated by representative DAPI staining images. The analyses were performed by comparing neo-formed tissue in the WM and the tissue of the NC.

The FT-skins were divided into three regions according to the wound gap position (wound bed, right edges, and left edges with respect to how the images are shown) and the orientation of the major axis of the nuclei of the basal cells in the epidermis was quantified. The dotted lines in Figure 3A_1_,A_2_ indicate the dermal–epidermal junction while the white arrows indicate the nuclei shape modified in the wound bed of the skins. On day 3 after the injury, a distribution that significantly differed from rotation angles of nuclei in unwounded tissues had been observed in the regions of wounded skin. In particular, nuclei-shape deformation increased moving from the unwounded lateral edges to the newly formed tissue. Nuclei orientation quantification is reported in Figure 3B, revealing a preferential cell distribution (90% of total number of cells) in the wound bed of the WM compared to the NC; unwounded tissues show a distribution of nuclear orientations with a tendency towards 90° (Figure 3B_1_), and wounded tissues towards 0–45° (Figure 3B_2_), when considering a Cartesian reference system where the X axis corresponds to the basal membrane (Figure 3B_3_).

#### 3.2.3. Gene Expression Results: KGF and TGF β-1 Expression in WM at Day 3

The expression of the KGF gene, as a crucial wound repair factor, and the TGF β-1 gene, as a regulator of ECM biosynthesis, was investigated. The analyses were performed by comparing the WM and NC tissues. As reported in Table 1 on day 3, a significant over-expression of both genes was found, reaching the RQ = 23.95 for KGF and RQ = 2.40 for TGF β-1, compared to the untreated control (NC = 1), suggesting, as expected, a strong pro-inflammatory response stimulating epidermal growth.

#### 3.2.4. Histo-Morphological Evaluation of Wound Model at 1–3–8 Days

Further investigation was performed by comparing the tissue morphology of the wound closure in the WM tissues, and the repair dynamics were investigated by H&E staining (Figure 4A–C) from day 1 to day 8. The epidermis seamlessly attached to the dermis compartment, except in the wound area on day 1 (Figure 4A), providing an in vivo-like matrix for re-epithelialization, surrounded by clearly defined wound margins. The tissue continuity was destroyed in both the epithelial and dermal compartment. The extending epidermal tongue is visible as 1–2 cell layers in the wound bed on day 3 (Figure 4B); the presence of cells randomly distributed in the wound bed, corresponding to the injury site, suggest the formation of a de novo tissue. The black arrow and the dotted line indicate the shift from the pre-existing tissue to a newly formed tissue. The keratinocytes covered 100% of the wound area on day 8 (Figure 4C), by organizing the epithelium into approximately three cell layers near the wound edges and one cell layer in the center of the wound bed. The dotted line indicates the boundary between the epithelium and dermis; keratinocytes are still visible in the wound bed.

#### 3.2.5. Keratinocytes Differentiation During Re-Epithelialization in WM: 1–3–8 Days

The keratinocyte differentiation in the wound area was investigated by IF for CK14 (green) and CK10 (red), by localization and expression in the newly formed epidermis within the wound bed at day 1, 3, and 8 in the WM (Figure 5A–C). The early expression of both CKs in the wound edge (Figure 5A) is visible: CK14 (green signal) is structured in the upper part (with respect to the white dotted line) while a signal featured by high background noise was observed at the lower part of the wound edges. CK10-positive cells are differentiated keratinocytes of the remaining tissue near to the wound bed. CK14-positive cells filled 100% of the wound bed within 3 days (Figure 5B), but the epithelium formed by these keratinocytes appears unorganized. A high background noise of CK10 was observed which emphasizes the presence of basal keratinocytes in the wound bed. On day 8 (Figure 5C), keratinocytes stratification in different layers is better and clearly observed; a signal of CK14 is well organized and is mainly localized in the basal layer of the multilayer epithelium. CK10 begins to be expressed appearing in the upper layers and it is directly related to keratinocytes differentiation.

#### 3.2.6. Fibroblasts Activation and New ECM Components in the Wound Bed: 3–8 Days

The dermis compartment in the wound area was investigated by (i) fibroblast differentiation and (ii) new ECM production on day 3 and day 8 in the WM. Figure 6A,B show the expression pattern of α-SMA (marker of myofibroblast differentiation typical of fibroblast heterogeneity) and of FN (specific biomarker of granulation tissue) by quantitative IF analyses.

In Figure 6A, the results obtained in the WM on day 3 and day 8 are shown; α-SMA-positive fibroblasts are randomly identified on day 3 (Figure 6A_1_,A_2_), while only a signal featured by high background noise of FN was observed (Figure 6A_1_). On day 8 (Figure 6B_1_,B_2_), α-SMA-positive fibroblasts are mainly located around the tongue of the newly formed epidermis. Moreover, in the wound area, the FN expression levels increase around the α-SMA-positive cells (Figure 6B_1_), and the spotted signal of α-SMA-positive fibroblasts corresponds to a strong red signal in the region of the wound.

In Figure 6C, the quantification of the cell density and the regulation of the α-SMA expression levels in the WM is reported. The expression levels of α-SMA in the WM are 30 and 35 times higher than the control (NC) on day 3 and day 8, respectively (*p* < 0.01); a positive fibroblast population corresponds to 30% and 35% of the whole cell population.

Figure 6D reports the quantitative analysis of the FN expression in the WM on day 3 and day 8.

The FN expression levels increased significantly on day 8 for both models. The statistical difference obtained from day 3 to day 8 (*p* < 0.01) highlights a time-dependent ECM production.

### 3.3. Results of Endolysin XZ-700 Efficacy

#### 3.3.1. *S. aureus* Immunolabeling (Cross-Vertical Sections and Top-View Acquisitions)

The evaluation of the *S. aureus* growth and adhesion was performed by a histological analysis by cross sections and top-view images, after 3 h from colonization for 24 h of treatment with XZ-700 compared to the placebo. Figure 7 shows representative images acquired at 10X magnification, while the signal quantification is reported in Figure 8.

In all colonized series, *S. aureus* was distributed in the injured area and over the stratum corneum (SC) surface (close to the lesion), partially aggregated in clusters (Figure 7, INJURED + *S. aureus*, slide 1 and 2). Treatment with XZ.700 300 µg/g resulted in a significant reduction in *S. aureus* adhesion/distribution along and within the tissue sections (* *p* < 0.05) compared to the colonized control. No effect on *S. aureus* reduction was observed in vertical cross sections for the placebo series (XZ.700 0 μg/g). These results were confirmed by signal quantification (Figure 8).

The bacterial distribution and adhesion on wounded tissues was investigated on the top-view acquisition in order to evaluate the effect of the product on the microbial adhesion and colonization from a different point of view. Figure 9 shows the most representative images of the 3 series.

The signal quantification of *S. aureus* positive microbial cells is reported in Figure 10. As expected, the untreated series was severely positive for the *S. aureus* signal (Figure 9, INJ + *S. aureus*), suggesting that the microbial infection diffused after the colonization on the surface around the lesion and inside the wound (cross section IF). The anti-adhesion effect of XZ.700 300 μg/g was also confirmed by the signal quantification of the top-view acquisitions (Figure 10); values of the pixel sum intensity significantly decreased, suggesting a direct effect on the microbial colonization capacity (and tissue penetration). Exposure to the placebo (XZ.700 0 μg/g) did allow a non-significant reduction in the microbial load on the epithelial surface, around the lesion (yellow circle), suggesting a limited early effect on the bacterial invasion.

#### 3.3.2. Scanning Electron Microscopy Analysis

Ultrastructural analysis performed by SEM allows deep investigation into the direct effect on biofilm *S. aureus* formation and the modulation of microbial adhesion on damaged FT-skin tissues treated both with XZ.700 and its placebo. Figure 11 shows the most representative images.

SEM showed an FT-skin surface with a homogeneous colonization and well-defined bacterial clusters in the replicative phase and in the early (not yet organized as biofilm) phase of PIA production (Figure 11A, green arrows).

In the treated series, *S. aureus* appeared dysmorphic or microcytic and its morphological modifications could be associated with product efficacy (Figure 11B, green arrows). Moreover, *S. aureus* was organized in small aggregates and distributed randomly. On the contrary, a dense and structured biofilm was visible with viable and strongly adherent bacteria (Figure 11C, green arrows) on the placebo series.

## 4. Discussion

To follow the main sequential steps involved in the re-epithelialization and wound healing process and to monitor how cellular and extra-cellular processes participate in the whole repair process, a standardized and reproducible injury procedure has been developed on an FT-skin model. The use of a punch (that removes the deep wounded tissue) ensured a reproducible and reliable lesion, by destroying the basement membrane and severely perturbing the cell–cell interaction. The disruption of the cell–cell interaction is necessary to enable keratinocytes to induce first the migration and second the proliferation and differentiation from the wound edge over the damaged area [7,54]. The tissue response to the injury has been monitored from day 0 to day 8 after the injury, with reproducible results on three independent experiments in terms of morphological modifications associated with specific proteins’ expression and localization.

The wounded FT-skin model (WM) was able to recapitulate the main steps of the wound healing process in the two skin compartments (epidermis and dermis). First, the migration of keratinocytes was confirmed by analyzing the CK14 and ITG-β1 expression and localization in the wound bed (Figure 2) and the potential cell polarization (Figure 3). The results obtained agree with the data in the literature, confirming cell migration as the first mechanism involved in wound re-epithelization [15,55,56,57,58].

The second mechanism involves cell proliferation and differentiation, which is mediated by the expression of specific CKs [55]. Indeed, the specific, localized, and time-dependent expression of specific CKs suggests the formation of a multilayered epithelium after the disruption of epithelia continuity by differentiating from basal (CK14-positive cells) to supra-basal (CK10-positive cells) keratinocytes [17]. We hypothesize that cells in the margin of the wound gap start with an initial sheet-like movement of individual epithelial cells (up to 3 days), involving both basal and supra-basal cells. This individual cell migration is followed by a landslide-like movement of the remaining epithelium that proliferates and differentiates to rebuild/remodel the architecture of the tissue as observed in Figure 5 [58].

To investigate ECM remodeling as a delayed healing mechanism, fibroblast localization and activation and α-SMA and FN expression have been addressed. In the WM, the overexpression of KGF and TGF-β1 has been quantified on day 3 (Table 1). The role of fibroblasts-differentiated myofibroblasts, as the main actor of the ECM remodeling in tissue repair in an organotypic model, and the importance of FN expression during the repair is well known [59,60,61]. Myofibroblasts are associated with many phenomena, such as wound contraction, closure, granulation tissue formation, and new ECM [60,62,63,64],while FN overexpression is associated with fibrosis and scarring [18,65,66]. The role of dermal fibroblasts in an open wound starting to proliferate from the wound margin one week after an injury has been widely reported. After re-epithelialization, a granulation tissue, essentially composed of HA (hyaluronic acid), FN, and fibroblastic cells that are activated and modulated into myofibroblasts, replaces the provisional matrix in the wounded connective tissue [61]. Consequently, the presence of myofibroblasts is fundamental for connective tissue repair [58].

According to the results shown in Figure 4, Figure 5 and Figure 6, the WM responded to the injury stimulus by both activating cells and stimulating the production of new ECM in the repaired tissue. The WM is able to re-epithelialize and the fibroblasts are the main population of the ex novo dermis compartment except for the wound gap area. This observation agrees with a study conducted by Lombardi et al., which also showed the appearance of cell differentiation and an increase in the levels of FN in an organotypic wounded dermal model [67].

The results obtained on the wounded FT-skin confirm that a reproducible and biologically relevant wound was established for as long as the adopted experimental conditions mimicked, from the very beginning (1–3 days), the key events involved in re-epithelization (8 days) [58].

To mirror complications of specific skin lesions, an infected and stable wounded model established on a fully viable FT-skin was produced by colonizing the wound bed with *S. aureus* for 24 h, as it nicely mirrors the bacterial–skin interaction that occurs in specific skin disorders.

Considering the specific mechanism of action of XZ.700 300 μg/g endolysin, a short experimental window (24 h) was suitable to assess its efficacy in counteracting *S. aureus* growth, adhesion, and biofilm formation. From 3 h after the colonization, as the time required for bacterial adhesion and proliferation, the tissues were treated with XZ.700 300 μg/g and with the placebo (XZ.700 0 μg/g) for an additional 24 h. XZ.700 showed strong antimicrobial activity (X log-reduction of *S. aureus*), suggesting a direct enzymatic effect of the test item in counteracting the molecular and chemical interaction between the bacterium and epidermal cells. An ultrastructural analysis by SEM confirmed the direct effect of the test item on the bacterial colonies’ aggregates; microcytic colonies were visible, and morphological modifications affected the microbial capacity of invasion, proliferation, and biofilm formation. It is interesting to notice that in the series treated with the placebo, *S. aureus* has probably taken an advantage of the components of the placebo formulation to form aggregates that better adhere to the FT-skin surface (top-view analysis).

## 5. Conclusions

The experimental model developed on FT-skin recapitulates, in a reduced time course, the fundamental steps of the skin regeneration process in the two skin compartments. It has been demonstrated to be a relevant, robust, and reproducible preclinical in vitro model, that spontaneously showed an intrinsic physiological capacity of tissue regeneration, starting the wound healing mechanism in 3 days for a total re-epithelialization in 8 days. This model could be applied to investigate the skin regeneration properties of topically or systemically applied ingredients and formulations mirroring real-life exposures and doses.

The in vitro wound model on FT-skin was successfully colonized with *S. aureus* to mirror the infected wound microenvironment during 24 h, in order to appreciate the efficacy of endolysin that acts by peculiar mechanisms of action on bacterial infections: (i) it works as an antimicrobial enzyme that targets the peptidoglycan layer of bacteria with a high specificity compared to antibiotics (avoiding the antibiotics-resistance phenomena), (ii) it has a rapid mode of action (bactericidal activity within a few minutes), and (iii) it causes negligible skin penetration due to the larger molecule size [49]. The results of this research project have confirmed that endolysin in gel formulation has maintained its efficacy in counteracting *S. aureus* growth and biofilm formation, avoiding complications occurring during wound healing processes and indirectly facilitating re-epithelialization. Further analyses are underway to evaluate the possible additional contribution of XZ-700 on re-epithelization activity in terms of tissue response and repair times.

The described preclinical in vitro infected wound model on FT-skin also has other potentially interesting applications to assess the wound healing efficacy of dermo-pharmaceutical and cosmetic formulations at doses and exposures mirroring real-life use, not only on the key steps of wound healing, but also to explore the contribution of the skin microbiome and in particular of pathogens, aiming to deeply investigate unknown MoAs in molecular and chemical signaling involved in physiological and pathological conditions.

## Figures and Tables

**Figure 1 pharmaceutics-16-01539-f001:**
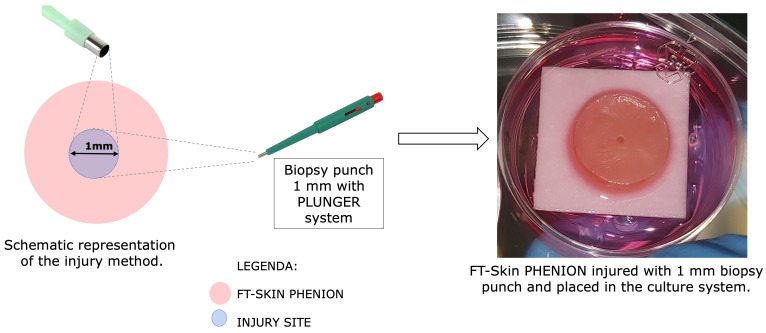
Schematic representation of injury procedure and macroscopic view of wounded FT-skin.

**Figure 2 pharmaceutics-16-01539-f002:**
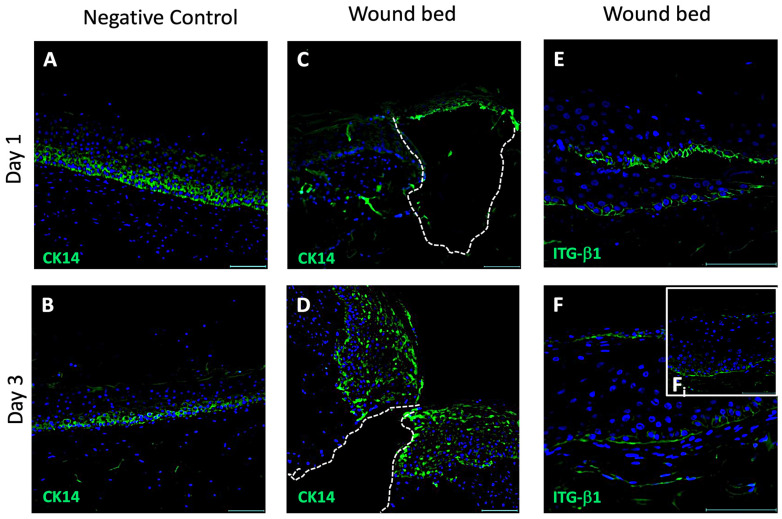
Immunofluorescence on CK14 and ITG-β1 in the WM. Representative fluorescent images from three independent experiments of CK14 in negative control (**A**,**B**) on day 1 and day 3, respectively, and of CK14 (**C**,**D**) and ITGβ1 (**E**,**F**) in the wound bed in the WM on day 1 and day 3, respectively. The wound bed is underlined by dotted lines. The insert Fi shows the right edge of the WM on day 3. Nuclei are stained with DAPI (blue). Magnification 20X (**A**–**D**); Magnification 40X (**E**,**F**); Scale bar: 100 μm; WM: wounded skin model; CK14: Cytokeratin 14; ITG-β1: Integrin-β1.

**Figure 3 pharmaceutics-16-01539-f003:**
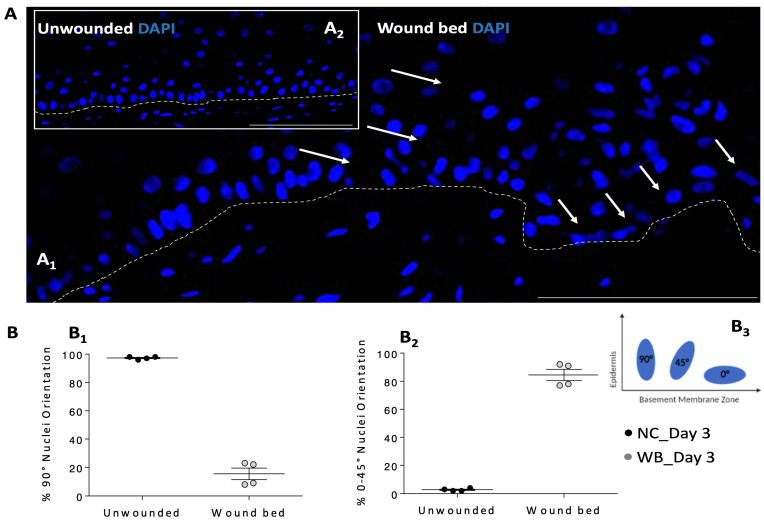
Nuclei orientation in the unwounded tissue (NC) and in the wound bed of the WM. Data are generated from three independent experiments. (**A**) Histological representation of nuclei stained with DAPI (blue) in wound bed (**A_1_**) and in the NC (**A_2_**) on day 3. White arrows indicate nuclei deformation. Dotted lines indicate dermal-epidermal junction. (**B**) Nuclei orientation (90° and 0–45°) in NC and wound bed. Unwounded tissues show a distribution of nuclear orientations with a tendency towards 90° (**B_1_**) and wounded tissues towards 0–45° (**B_2_**). A Cartesian reference system was used where the X axis corresponds to the basal membrane (**B_3_**). Magnification 40X. Scale bar: 100 μm. WM: wounded skin model, NC: negative control, WB: wound bed.

**Figure 4 pharmaceutics-16-01539-f004:**
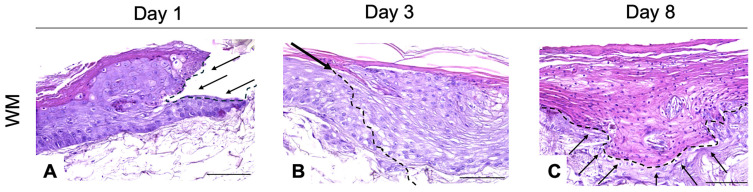
H&E on WM. Data are generated from three independent experiments. H&E stain shows morphological variation and progressive closure of wounds in WM day 1 (**A**–**C**). The black arrow and the dotted line in (**A**) indicate the shift from the pre-existing tissue to a newly formed tissue. The dotted line in (**C**) indicates the boundary between epithelium and dermis. Magnification 40X. Scale bar: 100 μm. H&E: Hematoxylin and Eosin; WM: wounded skin model.

**Figure 5 pharmaceutics-16-01539-f005:**
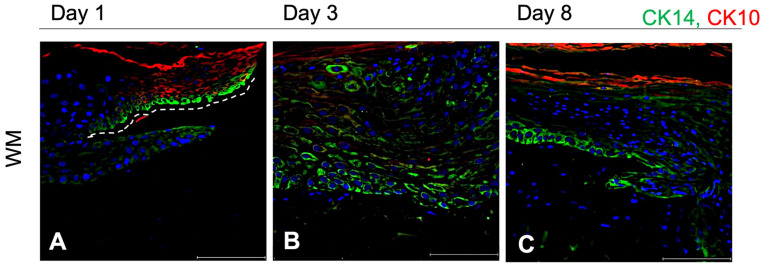
IF for CK14 and CK10 on WM. Data are generated from three independent experiments; the figure includes representative fluorescent images of CK10 (red) and CK14 (green) double staining of the wounded region in the WM on day 1 (**A**), day 3 (**B**), and day 8 (**C**), respectively. CK10 is used as differentiation biomarker while CK14 is used as marker for undifferentiated basal keratinocytes. Nuclei are stained with DAPI (blue). Magnification 40X. Scale bar: 100 μm. CK10: Cytokeratin 10; CK14: Cytokeratin 14; WM: wounded skin model.

**Figure 6 pharmaceutics-16-01539-f006:**
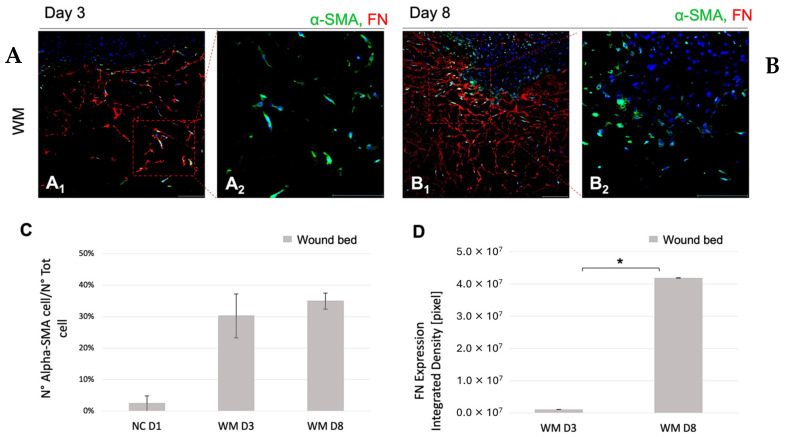
IF for α-SMA and FN on WM and signal quantification at 3 (**A_1_**,**A_2_**) and 8 (**B_1_**,**B_2_**) days. The figure includes representative fluorescent images of FN (red) and α-SMA (green) in the wounded region on day 3 and day 8, respectively. Nuclei are stained with DAPI (blue). Magnification 20X. Scale bar: 100 μm. (**C**): percentage of cells positive to α-SMA with respect to total number of cells. (**D**): quantification of FN Expression. * *p* < 0.05.

**Figure 7 pharmaceutics-16-01539-f007:**
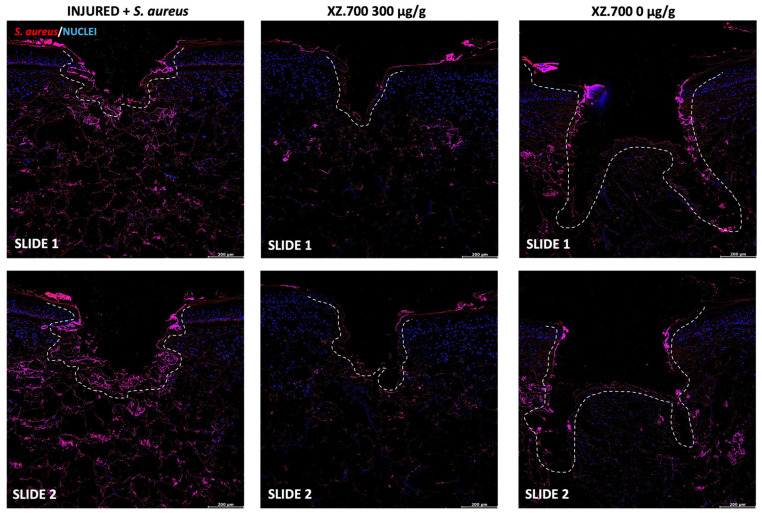
*S. aureus* immunofluorescence staining performed on wounded and treated paraffin sections. Dotted lines indicated the wound boundaries. Two slides were immunolabeled for each condition (biological simplicate). 10X Magnification. Scale bar: 200 μm.

**Figure 8 pharmaceutics-16-01539-f008:**
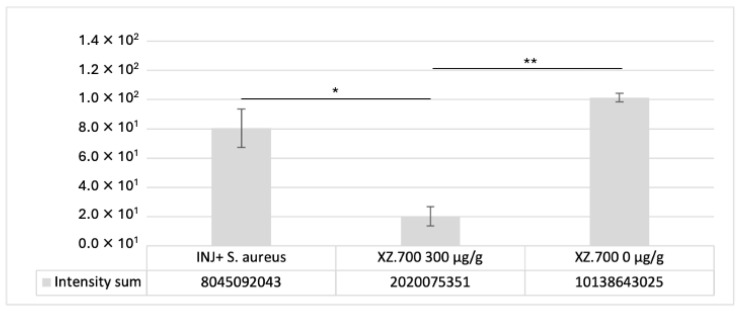
*S. aureus* quantification on paraffin cross sections, slides mean (biological simplicate). The signal was expressed as intensity sum. Statistical analysis performed by ANOVA T-test post-hoc Tukey; (* *p* < 0.05), (** *p* < 0.01).

**Figure 9 pharmaceutics-16-01539-f009:**
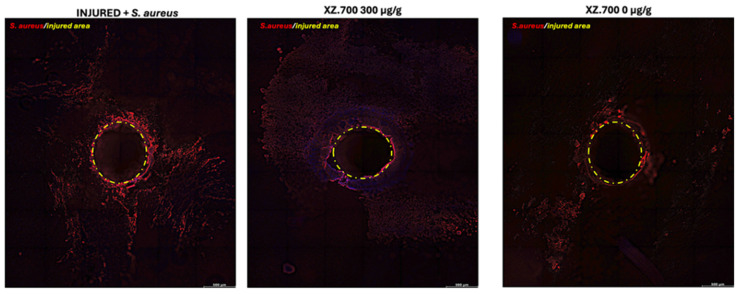
*S. aureus* immunofluorescence staining performed on wounded, top-view visualization. Tile Scan acquisition tool. Scale bar: 500 μm.

**Figure 10 pharmaceutics-16-01539-f010:**
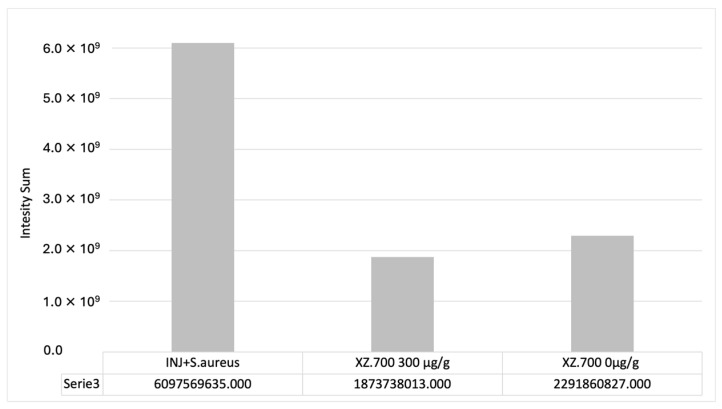
*S. aureus* quantification on top-view tissues (biological simplicate). The signal was expressed as intensity sum.

**Figure 11 pharmaceutics-16-01539-f011:**
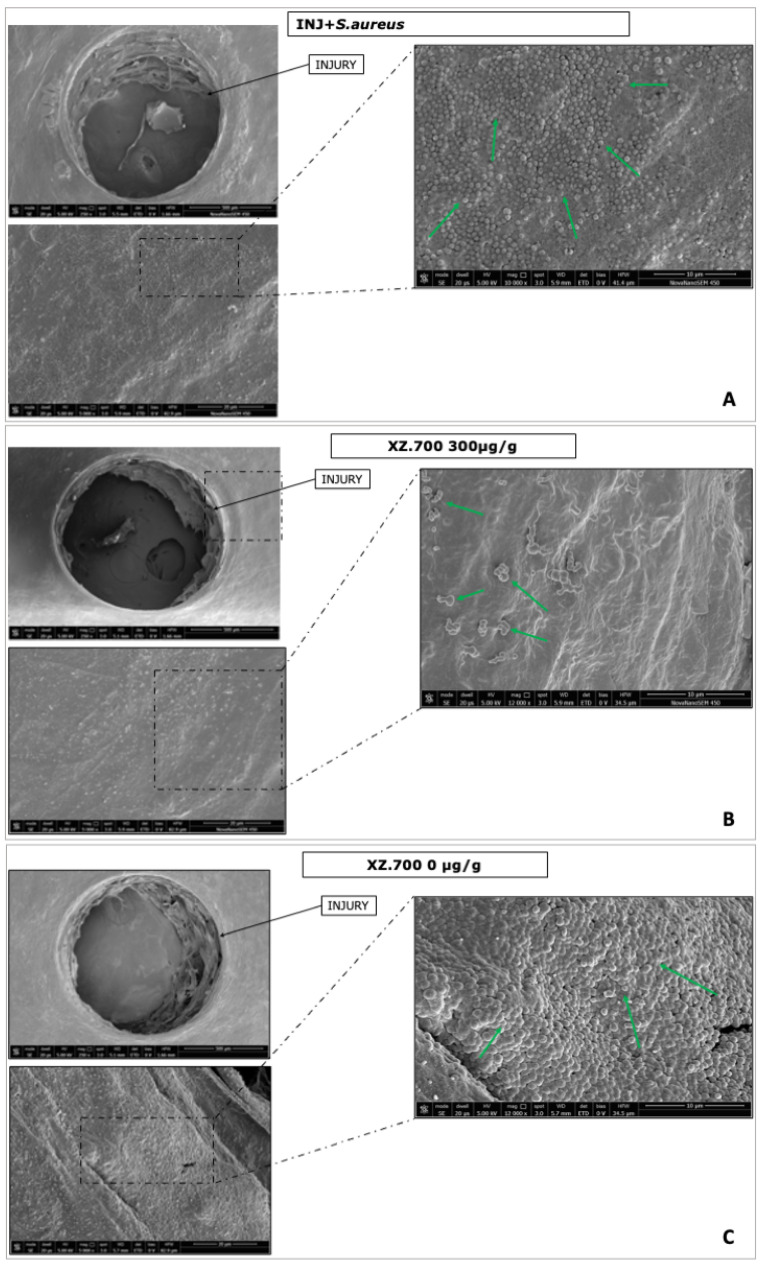
Scanning electron microscopy (SEM) acquisitions performed on colonized FT-skin. (**A**), FT-skin treated with XZ.700 300 µg/g (**B**), and FT-skin treated with XZ.700 0 µg/g (**C**). Magnification: 250X, 5000X, and 12,000X. Scale Bar: 10, 20, and 500 µm.

**Table 1 pharmaceutics-16-01539-t001:** Quantitative reverse transcription PCR analysis. qRT-PCR results of selected gene are reported for the Wound Model (WM) considering RQ values for the negative control (NC) at day 3 (D3_NC) = 1. KGF: Keratinocyte Growth Factor; TGF-β1: Transforming growth factor beta 1; RQ: relative quantification.

SERIES	KGF	TGF-β1
D3_WM	23.95	2.40 ^1^

^1^ RQ Study results compared to D3_NC as calibrator (RQ = 1.00).

## Data Availability

The original contributions presented in the study are included in the article, further inquiries can be directed to the corresponding author.

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
