# Peer review of "Targeted Antibacterial Endolysin to Treat Infected Wounds on 3D Full-Thickness Skin Model: XZ.700 Efficacy"

_pharmaceutics, 2024, doi:10.3390/pharmaceutics16121539_

Round 1
Reviewer 1 Report
Comments and Suggestions for Authors
The authors in the work entitled: Targeted antibacterial endolysin to treat infected wounds on 3D Full Thickness skin model: XZ.700 efficacy " aimed to develop an optimized in vitro experimental model using biological and molecular characterisation methods.
Overall this is a very nice manuscript with concise, coherent, and professionally presented results.
I only have a few questions:
1. Why did the authors choose Staph aureus ? Was there a possibility to use a gram negative bacteria as well?
2. Have the authors considered or is there an appropriate positive control for the XZ. 700 300 efficiency i.e. an antibiotic or other antiseptic compound ?
3. Is the -SMA and FN expression enough to adress the e ECM remodeling? Can it be termed significant: Have the authors considered of alternative stains ? Sirius red or Masson trichrome for example ?
4. What are the disadvantages of this model ?
5. Minor spelling mistakes can be found for examples in the keywords would-healing instead of wound-healing
Comments on the Quality of English Language
Minor issues.
Author Response
Comment 1. Why did the authors choose Staph aureus ? Was there a possibility to use a gram negative bacteria as well?
Response 1. The choice to colonize with staphylococcus aureus derives from the need to reproduce the possible complications that often arise during the wound healing phase: in many cases the repair and re-epithelization difficulties are caused by secondary infections caused by this bacterial strain, causing severe infections that aggravate tissue damage and slow the natural healing process. being the main cause of infections in case of skin damage, we focused on this type of bacterium.
Comment 2. Have the authors considered or is there an appropriate positive control for the XZ. 700 300 efficiency i.e. an antibiotic or other antiseptic compound ?
Response 2. Since it is an innovative molecule with a specific molecular effect that acts on the adhesion properties of the bacterial wall to the tissue, it was not possible to introduce an adequate positive control that could imitate the same mechanism of action.
Comment 3. Is the -SMA and FN expression enough to adress the e ECM remodeling? Can it be termed significant: Have the authors considered of alternative stains? Sirius red or Masson trichrome for example?
Response 3. From numerous studies it is known that the process of wound repair and re-epithelialization follow a specific dynamic in which FN and a-SMA are directly involved: in particular: α-SMA is a marker for myofibroblasts, that play a key role in wound contraction. Myofibroblasts differentiate from fibroblasts during the wound healing process and are essential for closing the wound by contraction of cells. On the other hand, Fibronectin is a key component of the provisional matrix that forms at the wound site. This matrix provides a scaffold for cell migration and supports the initial stages of wound healing. At the same time, FN promotes the adhesion and migration of keratinocytes and other cells move across the wound bed to re-epithelialize the damaged area.
Additional stains as PicroSirius Red or Masson Trichrome were performed, but they are qualitative stain not able to deep explain the repair dynamics, the key inducers involved and their expression.
Comment 4. What are the disadvantages of this model ?
Response 4. The possible disadvantage of the model is that it is based on a standard repair dynamics evaluated on a commercially available reconstructed skin model which does not allow the evaluation of a personalized repair process, where the patient's age, previous therapies or additional conditions pathologies can influence the dynamics of the shelters.
Comment 5. Minor spelling mistakes can be found for examples in the keywords would-healing instead of wound-healing
Response 5. Correct typos of key words.
Reviewer 2 Report
Comments and Suggestions for Authors
In my opinion, manuscript has been divided into two different parts: 1) studying re-epithelization of wound skin model 2) treating infected wounds on 3D Full Thickness skin model. This referee do not understand the relationship between the study of re-epithelization of the skin model and treating the skin model with XZ.700. I would suggest to focus on only one thing and provide further data. There is one option in order to link both studies: It would be great to study the re-epithelization of the skin model in the presence of XZ.700.
Introduction mus be improved, there is a lack of information in the material and methods. Figures are poor quality and more studies must be developed in order to support data showed. In general it is very difficult to understand what is shown in figures and localize the wound and the healthy tissue. For example, figure 5, CK14 is found in the basal layer and CK10 is a little bit higher. This is not shown in this figure. Another example, Figure 7, it seems that there are not differences in terms of bacteria reduction between Z.700.300 and Z.700. 0.
Comments on the Quality of English LanguageEnglish must be improved along the manuscript
Author Response
Comment 1: In my opinion, manuscript has been divided into two different parts: 1) studying re-epithelization of wound skin model 2) treating infected wounds on 3D Full Thickness skin model. This referee do not understand the relationship between the study of re-epithelization of the skin model and treating the skin model with XZ.700. I would suggest to focus on only one thing and provide further data. There is one option in order to link both studies: It would be great to study the re-epithelization of the skin model in the presence of XZ.700.
Response 1: The manuscript is in fact in a certain sense divided into two parts, in which, however, the first and second are nevertheless correlated: the basic idea is to propose a model useful for evaluating the physiological processes of re-epithelization, which occur spontaneously in conditions homeostatic following an injury; and a case of a lesion infected with Staph aureus which represents the main bacterium responsible for the main complications that can occur following a wound. This is certainly a preliminary study which lays the foundations for further investigations relating to the evaluation of hospitalization processes in the presence of bacterial contamination and in the presence of the XZ-700 product.
Comment 2: Introduction must be improved, there is a lack of information in the material and methods.
Response 2: Introduction and material and methods are improved with more details.
Comment 3: Figures are poor quality and more studies must be developed in order to support data showed. In general it is very difficult to understand what is shown in figures and localize the wound and the healthy tissue. For example, figure 5, CK14 is found in the basal layer and CK10 is a little bit higher. This is not shown in this figure. Another example, Figure 7, it seems that there are not differences in terms of bacteria reduction between Z.700.300 and Z.700. 0.
Response 3: Figures and descriptions are improved with more details. Regarding the figure 7, the localization of both CKs is as expected. The high magnification was chosen to focus the attention on the wound edges.
Comment 4: English must be improved along the manuscript.
Response 4: English has been improved.
Reviewer 3 Report
Comments and Suggestions for Authors
It is difficult to correlate the work with the title.
If the authors developed a new model, they should discuss the earlier methodologies reported by several other authors on this skin model and how their work on this skin model is different.
There is no information about the new molecule, to what category it belongs, and why they decided to test it.
How did the authors decide on the concentrations of the tested compounds?
Author Response
Comment 1: It is difficult to correlate the work with the title. If the authors developed a new model, they should discuss the earlier methodologies reported by several other authors on this skin model and how their work on this skin model is different.
Response 1: Improvement of the state of art regarding the existing skin model.
Comment 2: There is no information about the new molecule, to what category it belongs, and why they decided to test it. How did the authors decide on the concentrations of the tested compounds?
The product belongs to endolysins family, a novel class of antimicrobials with a rapid killing kinetics,. In particular “XZ.700” is composed of functional domains from the bacteriophage endolysin Ply2638 and the bacteriocin lysostaphin designed to specifically target S. aureus.
So, more details on molecule, characteristics and defined concentrations have been included.
Round 2
Reviewer 1 Report
Comments and Suggestions for Authors
N/A
Author Response
Ok, thank you for your support and your interest.
Best regards
Reviewer 2 Report
Comments and Suggestions for Authors
To have a more complete study and include both studies, authors have to study the re-epithelization of the infected skin model in the presence of XZ.700.
Comments on the Quality of English LanguageEnglish is ok
Author Response
Comments 1: To have a more complete study and include both studies, authors have to study the re-epithelization of the infected skin model in the presence of XZ.700.
Response 1: Certainly the data relating to the effect of the XZ-700 on re-epithelialization would be a step to take into account to give completeness to the overall results. In fact, these are studies that we are conducting precisely for this purpose.
Despite this, we decided to put this first data together anyway because, with the results relating to re-epithelialization, it was possible to demonstrate that the model is capable of spontaneously completing the healing of a deep wound, activating all the physiological processes orchestrated by the epidermis and from the dermis by activating remodeling mechanisms. Once it was demonstrated that it was possible to mimic a wound model, the activity of the product was then studied which, as the only mechanism of action, is to act as an endolysin, exerting a specific activity on bacterial adhesion of the St. aureus through the mechanism of action.
The subsequent effects of endolysin on re-epithelialization do not represent a direct effect as the action of the molecule is specific and directed against the adhesion of the bacterial wall to the tissue.
For the another evaluation and other key points, in which way you suggest to improve references and material and methods? Regarding the references, we take into account all bibliografy related to the most known state of art. Regarding the materials and methods we added all details on materials and experimental procedures. For specific experimental protocols, we could not share all info due to our internal stringent confidentiality procedures.
Thank you so much for your interest and support.
Reviewer 3 Report
Comments and Suggestions for Authors
The manuscript can be accepted
Author Response

(The authors gave the same response as above.)
